# An alternative representation of Synthetic Aperture Radar images as an aid to the interpretation of englacial observations

Álvaro Arenas-Pingarrón<sup>1</sup>, Alex M. Brisbourne<sup>1</sup>, Carlos Martín<sup>1</sup>, Hugh F.J. Corr<sup>1</sup>, Carl Robinson<sup>1</sup>, Tom A. Jordan<sup>1</sup>, Paul V. Brennan<sup>2</sup>

<sup>1</sup>British Antarctic Survey, High Cross, Madingley Road, Cambridge, CB3 0ET, United Kingdom
<sup>2</sup>Department of Electronic and Electrical Engineering, University College London, Torrington Place, London, WC1E 7JE, United Kingdom

Correspondence to: Álvaro Arenas-Pingarrón (alvnga99@bas.ac.uk)

Abstract. Ground penetrating radar reveals subsurface geometry and ice stratigraphy that contains information about past and present dynamics of the cryosphere. Synthetic Aperture Radar (SAR) is a processing technique based on averaging the radar echoes received at multiple locations as the radar moves relative to the target. Due to this averaging, the directional properties of the back-scattering from the target received at these multiple locations are lost. We introduce an alternative representation of SAR images that preserves directional information encoded in its Doppler spectrum: the Doppler frequency accounts for the time-delay variation from the radar to the target. With this technique, called Red-Green-Blue Doppler Decomposition (RGB-DD), the Doppler spectrum of a SAR image is split into three equalised bands, each band representing a primary direction of arrival of the radar echoes. A primary colour is assigned to each band to allow joint representation in a single RGB image. We apply our representation framework to several datasets acquired with the British Antarctic Survey (BAS) airborne ice-sounding radar over three Antarctic ice streams. Compared to the standard SAR method that is based solely on the averaged intensity level, this method facilitates the enhanced interpretation of englacial features such as ice stratigraphy, crevasses, tephra layers, and along-flow transitions in strain-rate. The technique may be extended to other sensors and applications.

## 1 Introduction

10

Synthetic Aperture Radar (SAR) images are an important tool for planetary surveys (Moreira et al., 2013). When applied to the cryosphere, and depending on the electromagnetic (EM) frequency band and acquisition method, SAR systems are used to observe processes and features at the surface, base and within the ice column. Surface measurements include elevation (Wingham et al., 2006) and ice flow velocity (Mouginot et al., 2019). Englacial features include firn stratigraphy (Panzer et al., 2013), ice sheet anisotropy (Dall, 2010), and ice thickness (Morlighem et al., 2013), with both 2D (Wang et al., 2016) and 3D imaging methods (Paden et al., 2010; Wu et al., 2011; Nielsen et al., 2017). SAR provides broad areal coverage, dependent on the platform carrying the radar. The platform may be a satellite (Wingham et al., 2006), an aircraft (Dall et al., 2010), a drone, or a surface vehicle (King et al, 2009; Paden et al., 2010). In active mode, the SAR both transmits and receives the signal, whereas in passive mode, the SAR receives a signal from a different source. Unlike a fixed antenna array, SAR systems

give constant along-track resolution, independent of the range (Moreira et al., 2013). This is achieved by increasing the number of radar locations used to focus deeper targets. The same resolution could be achieved with a large, fixed array of antennas if it was deployed with the required separation between the antennas.

SAR image interpretation relies on the reflectivity estimated from each target (scatterer) of interest (Schroeder et al., 2015; Holschuh et al., 2020). For airborne or spaceborne platforms, the EM propagation within the ice column is firstly controlled by the refraction at the air-ice interface, with Snell's Law limiting the incidence angle within solid ice to 34°. In addition, the echo reflectivity depends on the signal propagation losses within the ice column, as well as the properties of the target. For example, internal layering in continental ice sheets is different to that of fast-flowing areas at the ice sheet margins (Holschuh et al., 2014; Winter et al., 2015). Here, we present a new method that aids glaciological interpretation, introducing an additional imaging representation based on the direction of arrival of the radar echoes. Our method can be extended to other sensors, and non-glaciological targets, as well as by application to the EM spectrum and not only the Doppler frequencies.

## 2 Data and Methods

In this study we use data from the airborne SAR PASIN2 (Polarimetric Airborne Scientific INstrument, mark 2), a deep-ice sounding radar designed and operated by the British Antarctic Survey (BAS) for the 3D-mapping of the basal interface of the ice sheets (Arenas-Pingarrón et al., 2023b), and for polarimetric observations. An array of 12 antennas is mounted on a Twin Otter aircraft, with a non-linear and non-uniform distribution imposed by the aircraft structure. This antenna distribution makes PASIN2 an array with diversity in three joint domains: spatial distribution, across-track angular orientation (due to the subarray orientations), and across-track angular resolution (due to the distance between the sub-array elements).

#### 50 **2.1 Surveyed regions**

We present data from three Antarctic ice streams (Fig. 1a) over two different campaigns: Sentinel and Recovery ice streams in 2016/17; and Rutford and Thwaites glaciers in 2019/20. With PASIN2, the along-track time units are intervals called *traces*, with each trace having a duration of 200 or 400 ms respectively for the 2016/17 and 2019/20 campaigns. We employ trace numbers instead of distance to more easily enable replication of our work, as distance depends on the aircraft speed, which is variable, whereas traces have a constant duration. The SAR images and maps are labelled in *kilotraces* (ktr). With a nominal speed (v) of 60 m s<sup>-1</sup>, the distance for one kilotrace would be 12 km or 24 km, respectively for the 2016/17 and 2019/20 campaigns. Figures 1b-d present the flightpaths, with the thicker segments highlighting the data sections presented in this article.

Figure 1. Location of flightpaths used in this study. (a) Locations of campaign sites in Antarctica; (b) Track T04 over Rutford Ice Stream (400 s ktr<sup>-1</sup>); (c) Flight F26 over Sentinel Ice Stream (200 s ktr<sup>-1</sup>); and (d) track T11 over Thwaites Glacier (400 s ktr<sup>-1</sup>). The thick segments highlight the sections of data presented here. The black arrows indicate the ice flow direction. The along-track time units are *kilotraces* (ktr) with white dots marking intervals of 0.5 ktr in b and 1 ktr in c-d. The background colour is the surface flow speed (m yr<sup>-1</sup>) from MEaSUREs (Rignot et al., 2011; Mouginot et al., 2012; Rignot et al., 2017); the dashed black line is the grounding line from BEDMAP2 (Fretwell et al., 2013); and shaded surface reflectance is from LIMA (Bindschadler et al., 2008). All maps were generated using Quantarctica (Matsuoka et al., 2018; Matsuoka et al., 2021).

# 2.2 Data acquisition

PASIN2 transmits at a carrier frequency  $f_c = 150$  MHz (vacuum wavelength  $\lambda_0 = 2$  m), with bandwidth  $B_w = 13$  MHz. Eight under-wing antennas switch between transmit (TX) and receive (RX), four in each of the two wings, and four RX-only antenna are in a pod below the fuselage (belly). In RX, the 12 antennas are independent, whereas in TX, the group of four antennas below each wing transmit simultaneously as an array. PASIN2 transmits linearly varying frequency pulses (chirps), at an effective pulse repetition frequency (*PRF*) of 156.25 (2016/17) or 125 Hz (2018/19).

# 65 2.3 SAR Doppler representation

SAR images are typically in two dimensions, each with a different resolution: 1) the time-delay (range delay) dimension, related to the distance from the radar to the target of interest, with a resolution depending on the spectral bandwidth of each

received echo; and 2) the along-track dimension, corresponding to the location of the moving platform, with a resolution depending on the Doppler spectrum. The Doppler frequencies represent the rate variation of the range delay from the radar to each scatterer, as the radar moves in the along-track dimension.

If the radar track is parallel to the air-ice interface, the incidence angle from the radar in air ( $\alpha_0$ ) and ice ( $\alpha_l$ ) (relative to the normal of the ice surface, and positive towards the aircraft front) and the Doppler frequency ( $f_D$ ), are related by (Heister and Scheiber, 2018)

$$f_D(\alpha_0) = \frac{2 \cdot v}{\lambda_0} \sin \alpha_0 = \frac{2 \cdot v}{\lambda_0} n_r \cdot \sin \alpha_I \tag{1}$$

with  $\nu$  the speed of the platform, and  $n_r$  the refractive index of solid ice  $(n_r = 1.78)$ , relating the angles  $\alpha_0$  and  $\alpha_I$  by Snell's Law. The maximum incidence angle in solid ice  $\alpha_{I,M}$ , reached for  $\alpha_0 = 90^{\circ}$  (towards the horizon), is

$$\alpha_{I,M} = \sin^{-1}\left(\frac{1}{n_r}\right) \approx 34^{\circ} \ge |\alpha_I| \tag{2}$$

The incidence angles and the Doppler frequencies are positive when the radar approaches the target, and negative when moving away from the target. The Doppler bandwidth must be constrained to the effective *PRF* to avoid along-track aliasing, limiting the angles to

$$|\alpha_0| \le \sin^{-1}\left(\frac{\lambda_0}{2 \cdot \nu} PRF\right), |\alpha_I| \le \sin^{-1}\left(\frac{\lambda_0}{2 \cdot \nu \cdot n_r} PRF\right)$$
 (3)

The along-track focusing concentrates the Doppler frequencies into the shortest spatial or temporal resolution interval. When this is achieved, the intensity of the SAR representation of the scatterer is maximised. Regardless of the algorithm used, focusing involves an integration, or average, after which only the output (average) is presented, thus omitting the directional variability of the input data. This variability is directly related to the anisotropic (non-uniform) angular backscattering within the incidence beamwidth, and therefore of importance at the interpretation stage.

Assuming a constant aircraft height and terrain elevation, and if the scatterers are along the aircraft trajectory, the incidence angle of the EM wave from radar to scatterer is found from the Doppler frequency by inverting Eq. (1). Thus, the angular-incidence response is represented by the Doppler spectrum. Therefore, with this method, the SAR image includes the phase of the complex numbers representing each pixel, not only the amplitude: the phase may be lost in the standard representation and archiving of the image.

## 2.4 RGB Doppler Decomposition

70

By moving along the radar track, echoes are received over a range of incident angles and associated Doppler frequencies from each scatterer. This forms an aperture which resembles a cone pointing downwards (Fig. 2a). As stated above, the standard

Figure 2. SAR illumination with full aperture and RGB Doppler Decomposition: (a) full illumination beam as a grey cone with vertical axis; (b) Doppler spectrum with grey envelope between the limits of the effective pulse repetition frequency (*PRF*), and coloured bands in red (R), green (G) and blue (B); (c) cones of the coloured bands; and representative colour weights of (d) specular horizontal, (e) specular steeply dipping, and (f) rough surfaces.

SAR image only displays the intensity of the coherently averaged incidence angles within this aperture. However, the SAR processing algorithm recovers the Doppler spectrum within the image (grey envelope in Fig. 2b). This spectrum can be divided into several sub-bands by a filter array, with each filter corresponding to different beams which resemble cones (Fig. 2c). To jointly compare the intensity within the beams, the chosen sub-bands are each assigned to one colour within a triplet of colours, for example red(R)-green(G)-blue(B). These are then overlayed into a full-colour image, which retains a representation of the incidence angles by considering their relative intensities. This provides additional information for interpretation, such as specular (Fig. 2d-e) and diffuse (Fig. 2f) backscattering. The algorithm is presented in Appendix A. We refer to this spectral decomposition in the Doppler domain as "RGB Doppler Decomposition" (RGB-DD). The *RGB* term represents the additive model of the colour formation and is independent of the triplet of colours. For example, colours other than R-G-B may be used to overcome colour-blindness (see Appendix A).

Here, the more negative, moderate, and more positive frequencies are assigned to the first, second and third colours (called primaries), respectively, which, for this triplet, correspond to R, G and B. Usually, the Doppler spectrum will be symmetrical around zero-Doppler, so R would be negative frequencies, G around zero, and B positive frequencies.

The number of sub-bands need not be restricted to three, and with an array of Doppler bandpass filters, several sub-beamwidth responses may be obtained. In a previous study, Heister and Scheiber (2018) introduce the extraction of orientation and angular backscattering properties from SAR images using 29 sub-bands rather than the three presented here. Due to the large number of sub-bands, their output variability may be assessed with statistics such as the variance (Heister and Scheiber, 2018), skewness and kurtosis.

## 3 Representation of RGB Doppler Decomposition

115 If a target with diffuse scattering (a rough surface) is perfectly focused, when summing all the available Doppler spectrum the

Table 1 Image parameters for the RGB Doppler Decomposition, with PRF = 62.5 Hz and Doppler bandwidth of 30 Hz

| Flight (season) | Ice stream | Mean speed              | Aperture in air | Aperture in ice |
|-----------------|------------|-------------------------|-----------------|-----------------|
| T04 (2019/20)   | Rutford    | 55.2 m s <sup>-1</sup>  | 31.6°           | 17.6°           |
| F26 (2016/17)   | Sentinel   | $58.6 \text{ m s}^{-1}$ | 29.7°           | 16.6°           |
| T11 (2019/20)   | Thwaites   | $50.8 \text{ m s}^{-1}$ | 34.4°           | 19.1°           |

three sub-bands will combine into a pixel that ranges from grey to white, the greater the intensity, the brighter or whiter the pixel (Fig. 2f). When a specular (mirror like) target is focused, only a narrow bandwidth is backscattered (Fig. 2d, e), and its RGB colour will be one of the primaries within the triplet that represents the direction of arrival. If a target is unfocused, it will be represented by an alternating rainbow pattern of primary colours. This representation aids the image interpretation as it distinguishes between specular and diffuse backscattering, differentiates the englacial slope in regions where the layering cannot be identified, discriminates the ice base from multiples of the air-ice interface, detects shallow crevasses, and provides information regarding the quality of the focussing.

120

As an example of the Doppler decomposition, we present an across-flow section from Rutford Ice Stream, West Antarctica (Fig. 1b and Fig. 3), from the ice stream centre towards the shear margin, ~25 km upstream of the grounding line. Figure 3a presents the standard (amplitude-only) SAR image with the amplitude normalised to its maximum (Arenas Pingarrón et al., 2023a). The horizontal axis corresponds to the trace number (upper labels), and to the distance (lower labels). The vertical axis is depth (km), with the ice surface at 0 km and the bed between 2.1 km and 2.4 km.

The SAR processing aperture is vertical (0°), with half the beamwidth pointing forwards, and half backwards. The along-track sampling of the output image is equal to the PRF (62.5 Hz). For the RGB Decomposition, the Doppler bandwidth ( $B_w$ ) is 30 Hz (less than the effective PRF of the image). The mean aircraft speed is v = 55.2 m s<sup>-1</sup>, and by inverting Eq. (1), the apertures in air and deep ice are constrained to beamwidths of 31.6° and 17.6°, respectively (see Table 1).

Figure 3. SAR representations of Rutford Ice Stream, West Antarctica (Fig. 1b and Table 1), season 2019/20, flight T04, transmitting from starboard array, and receiving from port antenna P1: (a) image with amplitude (dB) normalised to its maximum; b) RGB-DD with 10 Hz sub-bands centred at -10 Hz (R), 0 Hz (G) and +10 Hz (B); (c) intensity and colour of the strongest sub-band from (b); and (d) RGB-DD with overlapping sub-bands, producing yellow (Y) and cyan (C). The area within the white dashed border is heavily crevassed or tightly folded. Within the white solid border, zoomed in Fig. B1, the contrast of the deep layering is improved with RGB-DD. The image is from the ice stream centre (~7.7 ktr) towards the shear margin (~8.16 ktr) and the ice flow direction is out of the page (see Fig. 1b). The vertical axis is the depth relative to the ice surface.

The RGB Doppler Decomposition is shown in Fig. 3b, with the amplitude limits as in Fig. 3a. The colours encompass the full Doppler bandwidth ( $B_w$ ) with no overlap. That is, red, green, and blue, represent the output of filters with sub-bandwidths of 10 Hz ( $B_{sw,1} = B_{sw,2} = B_{sw,3} = B_w/3$ ), centred at -10 Hz, 0 Hz and +10 Hz respectively ( $f_{D,1} = -10 \text{ Hz}$ ,  $f_{D,2} = 0 \text{ Hz}$  and  $f_{D,3} = +10 \text{ Hz}$ ). In each pixel, the colours are combined after being weighted according to the filter output amplitude. By inverting Eq. (1), the angle limits in deep ice ( $\alpha_{I,i}$ ) for each colour are

$$\alpha_{I,i} = \sin^{-1}\left(\left(f_{D,i} \pm \frac{B_{sw,i}}{2}\right) \cdot \frac{\lambda_0}{2\nu \cdot n_r}\right), i \in \mathbb{N}, 1 \le i \le 3$$

$$\tag{4}$$

- resulting in -8.8° <  $\alpha_{I,1}$  < -2.9°, -2.9° <  $\alpha_{I,2}$  < 2.9°, and 2.9° <  $\alpha_{I,3}$  < 8.8° for the mean speed, although it varies locally (here the standard deviation of the aircraft speed was 1.4 m s<sup>-1</sup>, meaning a standard deviation of 0.07° and 0.22° respectively for 5 Hz and 15 Hz). Therefore, deep-ice incidence angle limits of Eq. (4) define sub-beamwidths of ~5.8°, a third of the total ice beamwidth.
- As an alternative simplified representation, rather than combining the three sub-bands, each pixel is represented by the intensity and colour of only the most intense sub-band, as in Fig. 3c. When a rough surface does not favour a unique sub-band (Fig. 1f), it is represented by a random variation of red, green and blue. Where the slope alternates rapidly, as over severely folded areas for example, rapidly alternating red-green-blue is observed.
- To improve the identification of frequencies, the three sub-bands can be overlapped. If the scatterer response lies in the overlapping region, it will be represented by the mixing of two band colours, resulting in yellow (Y) and cyan (C) when using the R-G-B triplet. This decomposition is presented in Fig. 3d, with the red, green and blue now representing the sub-bandwidths 12, 18 and 12 Hz ( $B_{sw,1} = B_{sw,3} = 0.4 \cdot B_w$ ,  $B_{sw,2} = 0.6 \cdot B_w$ ), centred at -9Hz, 0 Hz and +9 Hz, respectively ( $f_{D,1} = -f_{D,3} = -0.3 \cdot B_w$ ,  $f_{D,2} = 0$  Hz). Each of the five colours (three primary and two secondary) occupy a fifth of the processed Doppler bandwidth, 155  $B_w/5 = 6$  Hz, and ~3.5° sub-beamwidths.

## 4 Applications

#### 4.1 An aid to the interpretation of englacial structures and features

The example from Rutford Ice Stream presented in Fig. 3 highlights several ways that the interpretations of englacial structures and features are facilitated by the RGB decomposition over a standard SAR image:

1) In Fig. 3a, a standard SAR amplitude image, layers steeper than 8.8° result in dark areas in the image (a result of the 15 Hz Doppler limit). In the RGB-DD image (Fig. 3b), these steep (dark) layers are now represented by a colour, reflecting the underlying pattern in the Doppler spectra, highlighting features that are lost with standard SAR processing and images, and enabling a qualitative interpretation. The underlying data, i.e., the Doppler bands (RGB-

- DD output), may be analysed in a quantitative and automated manner to extract features such as surface slope and dip direction. Furthermore, with the three RGB-DD filters, different decompositions may be achieved by varying the centroids and bandwidths and combining the results to produce further bands, like those in Fig. 3d, allowing additional quantitative analysis and improved angular resolution.
  - 2) The contrast in the coloured images (Fig. 3b-d) is improved relative to the standard SAR image, because the sub-band equalisation (Appendix A) offers features visible to greater depths than in the SAR image. This is highlighted in Fig. 3 by the white rectangle, zoomed in Fig. B1 with the same amplitude scale as Fig. 3 for comparison.
  - 3) With Doppler decomposition (Fig. 3b), the bed is light grey or white (as R, G and B are uniformly combined), meaning the EM waves are backscattered over the full 17.6° ice beamwidth and indicating a rough surface.
  - 4) The shallow ice is bright green down to a depth of 300 m (Fig. 3c) indicating sub-horizontal layering, but towards the shear margin (8.02 ktr) tight folds are observed at a depth of ~180 m (within the region enclosed by the dashed white line). This is indicated by alternating red-green-blue pixels that represent alternating dip directions, extending to shallower and deeper ice approaching the margin.
  - 5) From ~8.06 ktr, the alternating red-green-blue becomes bright white at a depth of ~150 m, and as crevasses behave as corner reflectors backscattering the EM waves in all directions, these bright-white features are interpreted as crevassing extending towards the surface near the margin.

## 180 4.2 Identifying rapid increase in the surface strain-rate

Figure 4 presents the standard SAR amplitude and RGB-DD images from flight F26 over Sentinel Ice Stream on the English Coast of West Antarctica (Fig. 1c and Table 1; Corr et al., 2021). The flightpath follows the ice flow direction to the ice shelf and then turns to return up the ice stream, resulting in the symmetry of the image, with the central section capturing the ice shelf base and its multiple (respectively at -200 and -700 m, between ~59 and ~60 ktr). At traces 57.58 ktr and 61.77 ktr (identified by white arrows in Fig. 4b,c), the Doppler decomposition highlights two transitions, the first being abrupt, and the second more gradual. These locations are not associated with turns of the aircraft, but correspond with the transition to a flow speed >350 m yr<sup>-1</sup> (see Fig. 1c). From the on-board Lidar altimeter, we calculate the ice surface slope (measured from the horizontal) in the along-track direction (Fig. 4d), which highlights that these features in the RGB-DD representation occur at locations of rapid variation of the surface elevation, that are attributed to surface crevasses. Our hypothesis is that this along-flow variation of scattering, from specular to broad, is a consequence of the transition to higher strain-rate (which produces surface crevassing). This is highlighted in Fig. 4a-c within the solid rectangle, zoomed in Fig. B2. Although the features are present in the standard SAR image, they are more obvious in RGB-DD and the interpretation could be automated. Furthermore, surface crevasses may not always be present in Lidar data, for example, when buried.

Figure 4. SAR representations of Sentinel Ice Stream (Fig. 1c and Table 1), season 2016/17, flight F26, transmitting from starboard array, and receiving from port antenna P1: (a) standard SAR image with amplitude (dB) normalised to its maximum; (b) RGB-DD with sub-bands of 10 Hz centred at -10 Hz (R), 0 Hz (G) and 10 Hz (B); (c) intensity and colour of the strongest sub-band with triplet  $Y_D$ - $G_D$ - $B_V$ ; and (d) surface slope in the along-track dimension (reported in degrees from the horizontal), from LIDAR. In (b) and (c) the arrows mark the locations of along-flow transitions from specular (uniform colours) to broad scattering (white and random colours). The area within the white solid border, is zoomed in Fig. B2.

# 4.3 Identification of tephra layers

Tephra is volcanic material of varying size and composition that is ejected during volcanic eruptions. During a volcanic eruption, tephra will form a layer on the snow surface that will subsequently be buried and transported within the ice drainage basins, and therefore deformed and transported in the same manner as all other internal layers (Corr and Vaughan, 2008). Tephra is a direct indicator of volcanic activity, which can influence ice flow and stability, as well as a useful record of ice

Figure 5. Possible englacial tephra layers in SAR representations of Thwaites Glacier (Fig. 1d and Table 1), season 2019/20, flight T11, transmitting from starboard array, and receiving from starboard antenna SC: (a) image with amplitude (dB) normalised to its maximum; b) RGB-DD with sub-bands of 10 Hz centred at -10 Hz (red), 0 Hz (green) and 10 Hz (blue); and (c) intensity and colour of the most intense sub-band with triplet  $Y_D$ - $G_D$ - $B_V$ . The tephra layers are marked with white arrows. The section starts ~3.5 km away from the western margin (7.3 ktr) and covers ~7.9 km in the across-flow direction (until 7.7 ktr).

dynamics.

Figure 5 presents SAR and RGB-DD profiles from the western margin of the Thwaites Ice Stream near Mount Takahe volcano (Wilch et al., 1999) (Fig. 1d and Table 1) (Jordan and Robinson, 2021). We interpret several layers as tephra (Hoffman, A. O. (2022), pers. comm.), indicated by arrows in Fig. 5. The snow surface is at a height of ~1 km above ellipsoid but is unclear due to the surface echo arriving during signal transmission (during acquisition of this image the RX antenna was also part of the TX array). Three tephra layers are highlighted in Fig. 5 (between the along-track locations ~7.39 and ~7.64 ktr, at ellipsoidal

Figure 6. Zoom-in of Fig. 5, to show the upper interpreted tephra layer with white arrows. The broad scattering of the layer between 7.3 ktr and 7.45 ktr is only distinguished in (b). The black arrows are range sidelobes of the main response.

heights of 100 m and -700 m). Layers become whiter and thicker with depth, either through deformation or percolation of the tephra layer through the ice column. A deeper fourth layer is found from ~7.47 to 7.54 ktr, reaching the bed. At 500 m height, a short white section of a layer extends from ~7.63 to ~7.65 ktr (Fig. 6, right-hand white arrow). This section is brighter than the deeper layers, indicating a greater concentration of tephra. The bright scattering marked by the black arrows in Fig. 6 is produced by range sidelobes of -20 dB (replicas of the tephra feature); the range location of these sidelobes is currently unexplained.

Around 7.38 ktr, scattering is not visible in the standard SAR image (Fig. 6a) but is clear in RGB-DD (Fig. 6b). Figure 6c,

presenting only the most intense RGB band, is contradictory as the main scattering is greater from negative Doppler frequencies, in disagreement with the positive slope of the layer. This may be a result of the orientation of the internal structures or geometry of the tephra layer itself (Fichtner et al., 2025).

#### 5 Discussion

## 5.1 Comparison with standard SAR representation

Using RGB Doppler Decomposition of SAR data, we visually represent the directional scattering of englacial features. This is absent in standard SAR image representations. The directional scattering of the data presented here is mostly related to the slope of internal layers, which are not always traceable with standard SAR representations. With colour as an additional differentiation factor, our technique aids in the interpretation of SAR images and will improve automatic feature recognition and layer tracking. For example, in a conventional SAR image, a bright specular reflector (Fig. 2d) is indistinguishable from a rough reflector that results in a broad but moderate response (Fig. 2f). However, after combining the three sub-bandwidths, crevasses and tephra layers are white or light grey, whereas a bright ice layer will be coloured according to the slope.

The method will also aid in the differentiation of multiple-bounce surface returns. In RGB-DD, a rough bed is characterised by a light grey colour, whereas the smooth air-ice interface will be represented by the perpendicular sub-beam (typically near zero-Doppler or green in our standard colourmap, as in Fig. B2). In this manner, multiple surface returns may be distinguished from bed returns.

## 5.2 Comparison with other methods

Other techniques are available to measure the slope of internal layers, and these other methods will determine the slope with greater accuracy than the RGB-DD method presented here (Holschuh et al., 2017; Heister and Scheiber, 2018; Castelleti et al., 2019). However, some of these methods require optimization of the processing parameters during the along-track focussing, and they cannot be applied at either the range-compressed level or after the SAR processing. Doppler Decomposition may be applied without the full processing sequence applied and therefore provides more flexibility or even rapid application for quality control during data acquisition, for example.

RGB Doppler decomposition is similar to the method of Heister and Scheiber (2018) (referred to here as the multi-band method). The multi-band method employs Doppler filters equivalent to free-space sub-apertures of 2° beamwidth with 1° overlap, resulting in 29 filters for a 30° SAR-aperture. The main advantage of the multi-band method over RGB-DD is the enhanced angular resolution, which is five times better due to the 2° beamwidth compared to the 10° beamwidth of the RGB method. Conversely, the along-track spatial resolution of the RGB-DD method is five times finer than with the multi-band method due to each of the RGD-DD filters integrating a wider beamwidth. Also, the use of only three filters allows

representation of the three outputs simultaneously in a single RGB image. This single image may be used to interpret the angular anisotropy of the backscattering: in a pixel, the variance of the three filter outputs varies the RGB saturation, or luminosity. By using multiple filters, variance fluctuations can be determined that may not be identified in our framework. However, to represent the anisotropy using the multi-band method requires an additional image of the filter variance. The source of the variance is difficult to assess unless all the filter outputs are shown, which is impractical with 29 filters. For this reason, the RGB-DD technique results in a more suitable representation to aid image interpretation.

Furthermore, at radar wavelengths the internal layers of ice behave as specular reflectors, backscattering perpendicular to their slope. Thus, with large numbers of sub-bands, the response of an ice layer is mostly within a single sub-band, and the multiband statistics become redundant.

255

Therefore, although similar to the multi-band method, the RGB-DD method divides the angular information into three groups that, despite losing angular resolution, provides a direct way to jointly display directional and reflectivity information with colours and intensities, and offers improved spatial resolution.

### 5.3 Application at other SAR processing stages

The Doppler decomposition can be applied to range-processed data (before SAR or along-track processing). In this case the decomposition is a powerful tool for assessing the data acquisition, as at this processing stage there are no assumptions about the path travelled by the EM waves, which are needed for the SAR focussing. Applied to the SAR image, our method assists in evaluating the focussing quality: if the rough ice base has been adequately focussed, it will be light grey or white. However, if the image is defocussed, the resulting along-track dispersion will present as a sequence of the three primary colours.

### 265 5.4 Other applications and future directions

Spectral decomposition may also be applied to any radar EM frequencies and not only the Doppler frequencies. However, in the case of PASIN2, with 13 MHz bandwidth centred at 150 MHz (relative bandwidth 8.7%), the spectral variability is too small to be useful.

This requires three steps: 1) initial processing of the SAR image; 2) application of the the RGB Doppler Decomposition and appraisal of the nature of the scattering to determine the optimal along-track aperture; and 3) SAR re-processing (from the unprocessed raw data) with the optimal Doppler bandwidth. In the third step, the scatterers are not assumed to be simply point targets with homogeneous scattering, thus improving the signal to noise ratio compared to the case when considering full bandwidth.

The RGB-DD method may be used to distinguish along-track ambiguities, i.e., features that are mislocated within an image due to Doppler spectral aliasing. These features will appear as a sequence of the primary colours, whose thickness and colour order indicate, respectively, the distance and location relative to the radar, highlighting their mis-location.

#### 280 Conclusions

285

290

305

We introduce the RGB-DD framework as an enhanced representation of Synthetic Aperture Radar (SAR) images for interpreting englacial surveys. As an alternative to conventional SAR representations that only show the backscattered power, the RGB-DD method incorporates knowledge of the along-track angle of arrival of the received radar signals. The angles of arrival are divided into three beamwidths, and each assigned to a primary colour band. After overlapping the three bands, the response of each can be jointly observed with a single full-colour image. For interpreters with colour blindness, we suggest alternative primary colours.

We apply our RGB-DD method to three regions. Compared to the standard SAR amplitude representation, coloured images facilitate interpretation by representing directional backscattering information that is otherwise lost. Our method improves the visualization beyond just internal layering. It distinguishes between specular and broad backscattering to reveal otherwise unrecognised features such as shallow crevasses, tephra layers, and transitions from low to high strain rates. The use of colour offers an additional dimension to aid the automatic identification of these features. RGB-DD is a rapid and flexible technique that can be applied at any stage of the SAR processing and is readily extended to other radar systems.

# Appendix A. Workflow

The spectral sub-bands are chosen so that they encapsulate the bandwidth of interest, which may be different at each processing stage. At the raw data level, or after the range focusing, the Doppler Decomposition sub-bands may add up to the sampling frequency, i.e., the PRF, if a Doppler filtering has not been performed. After the SAR focusing, the sub-bands may add up to the Doppler bandwidth, which is usually smaller than the PRF. The bandwidth and central frequencies of each sub-band depend on the application. It is convenient to perform different analyses and compare the results, for example with contiguous, non-contiguous, or overlapped sub-bands. For the finest resolution, the sub-bandwidth and sub-band separation are one third of the processed Doppler bandwidth. Hence, the along-track spatial resolution is lower than in the SAR image.

According to the sub-band definitions, three box-car filters are applied to the SAR image (or raw data) in the domain of interest (Doppler in the case of RGB-DD). For each of the three outputs  $S_i$ ,  $1 \le i \le 3$ , the steps are:

1. To compensate the antenna and backscattering patterns, by normalising  $S_i$  to its own maximum (the option chosen for the results presented in this article), otherwise by normalising to the maximum of all three outputs.

- 2. To convert to the logarithmic scale (dB).
- 3. To set maximum and minimum limits (in dB) for equalising the RGB image and improving the contrast (Fig. B1).
- 4. To quantify the logarithmic scale linearly from 0 to 255 between the previous maximum and minimum limits.
- This procedure results in three outputs  $Q_i$ ,  $1 \le i \le 3$ ,  $0 \le Q_i \le 255$ , and each  $Q_i$  is coded with a primary colour. For each colour, its three RGB components are rational numbers within the interval [0, 1]. For the R-G-B triplet, the primary colours  $C_i$  are

$$C_1 = R = RGB_1(1, 0, 0)$$
  
 $C_2 = G = RGB_1(0, 1, 0)$   
 $C_3 = B = RGB_1(0, 0, 1)$ 
(A1)

with  $RGB_1$  the base for the three components within the interval [0,1]. For universal readability, (A1) should be avoided, and we suggest the dark yellow  $(Y_D)$  - dark grey  $(G_D)$  - violet blue  $(B_V)$  triplet for to avoid misinterpretation by users with colour blindness (Arenas-Pingarrón, 2025)

$$C_1 = Y_D = RGB_1(0.55, 0.55, 0)$$
  
 $C_2 = G_D = RGB_1(0.25, 0.25, 0.25)$   
 $C_3 = B_V = RGB_1(0.20, 0.20, 0.75)$  (A2)

15 The only condition for the selected primary colours is that their full combination result in a pure white colour W

$$\sum_{i=1}^{3} C_i = W = RGB_1(1, 1, 1)$$
(A3)

Once the primary colours are chosen, without any loss of generality, the spectrally decomposed image RGB\_SD is

$$RGB\_SD = \sum_{i=1}^{3} Q_i \cdot C_i \tag{A4}$$

The RGB image will have the same dimensions as before the decomposition, but now with pixels of 24 bits (8 per sub-band).

#### Appendix B. Image details

Figure B1 and Fig. B2 enlarge the content within the solid rectangles in Fig. 3 and Fig. 4, respectively. In Fig. B1b, the contrast is improved relative to the standard SAR image representation, because the sub-bands are equalised in amplitude, by normalising to their maximum. In Fig. B2 are shown the distinctive RGB-DD colours of the ice surface and crevasses.

Figure B1. Zoom-in of white rectangle in Fig. 3a-b, showing the enhanced contrast of the RGB-DD after sub-band equalisation.

Figure B2. Zoom-in of the white rectangle in Fig. 4a-b highlighting how the surface presents as green in RGB-DD and crevasses as bright white.

## Code availability

The software for the RGB Doppler Decomposition and the generation of the suggested colourmaps is openly available in the 325 BAS GitHub repository at <a href="https://github.com/antarctica/sar-rgb-spectral-decomposition.git">https://github.com/antarctica/sar-rgb-spectral-decomposition.git</a> and Zenodo <a href="https://doi.org/10.5281/zenodo.14962614">https://doi.org/10.5281/zenodo.14962614</a> (Arenas-Pingarrón, 2025).

## Data availability

To test the results of this study, the SAR image of Fig. 3a is openly available in NERC EDS UK Polar Data Centre at <a href="https://doi.org/10.5285/40c2f86b-1a02-4106-934a-42769682df66">https://doi.org/10.5285/40c2f86b-1a02-4106-934a-42769682df66</a>, dataset ID GB/NERC/BAS/PDC/01766, (Arenas Pingarrón et al., 2023a).

#### **Author contribution**

All authors contributed to the manuscript. AB, HC, and PB supervised the research; HC, CR, and TJ planned the campaigns and collected the data; AAP, HC, CR, and TJ curated the data; AAP, HC, TJ, and PB developed the conceptualization and methodology; AAP developed the software; AAP, AB, CM, HC, and TJ interpreted the data.

# 335 Competing interests

CM is an editor of TC. The other authors declare that they have no conflict of interest.

## Acknowledgments

We thank K.W. Nicholls for the comments and suggestions, and the Norwegian Polar Institute's Quantarctica package <a href="https://www.npolar.no/en/quantarctica/">https://www.npolar.no/en/quantarctica/</a>. This work has received funding from the NERC grant NE/L013444/1, project: Ice shelves in a warming world: Filchner Ice Shelf System (FISS), Antarctica. The 2016/17 data were collected as part of the NERC grant NE/L013770/1, project: Ice shelves in a warming world: Filchner Ice Shelf System (FISS), Antarctica. The 2019/20 data were collected as part of the BAS National Capability contribution to the NERC/NSF International Thwaites Glacier Collaboration (ITGC) program. We are grateful to Nick Holschuh, an anonymous reviewer and the editor for their considered comments that have helped improve this manuscript.

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
