# Peer review of "An alternative representation of Synthetic Aperture Radar images as an aid to the interpretation of englacial observations"

_EGUsphere, 2025_

## Referee Comment (RC1)

**Review of:** An alternative representation of Synthetic Aperture Radar images as an aid to the interpretation of englacial observations
**Submitted to:** The Cryosphere
**Reviewer:** Nicholas Holschuh

**General Comments:**

In this manuscript, Arenas-Pingarrón et al. present a novel data visualization strategy to aid in the interpretation of ice penetrating radar data. Following the work of Heister and Scheiber (2018), Arenas-Pingarrón et al. use information from the range-Doppler spectrum to decompose backscattering into three components: energy arriving from nadir, energy arriving from along-track positions ahead of the radar, and energy arriving from along-track positions behind the radar. By color-coding backscatter intensities from these three directions using red, green, and blue bands, it is possible to include specularity and directional scattering information in a traditional radar image, and aid in qualitative interpretation of subsurface features.

There are opportunities to improve the precision and clarity of the writing, most notably in the introduction. But beyond the writing, I have no major technical criticisms of the manuscript.

My primary concern is its fit in "The Cryosphere". As stated above, the main contribution of this work is a new data visualization strategy. The authors' approach to rendering range-Doppler spectral information is both novel and intuitive, but the underlying quantitative interpretation framework has already been presented by Heister and Scheiber (2018), which the authors acknowledge readily throughout the work. To me, these images would have their greatest value as an intermediate data product in the radar processing and interpretation work-flow; they would provide a useful quick-look reference to inform one's choice in focusing aperture (as the authors describe in section 5.3) and help interpreters narrow the geographic scope of more quantitative investigation of the range-Doppler spectra, likely following Heister and Scheiber. In that way, this manuscript seems more appropriate for a venue like IEEE Transactions on Geoscience and Remote Sensing, where the broader radar community might encounter it and be able to consider the utility of these images.

**Line-Item Corrections:**

| | |
|---|---|
| Page #: 1
Line #: 11-12 | The two sentences starting "Due to this averaging…" are unclear to me. What is a "directional feature"? What is the "distance rate from the radar"? Consider revising these for clarity. |
| Page #: 1
Line #: 24 | Your verb choice here makes me stumble as I read -- what does it mean to use radar to "address processes and features"? |
| Page #: 1
Line #: 28 | What do you mean when you say that coverage "depends on the trajectory of the moving platform"? Why specifically "the trajectory"? I'm not sure what point you are trying to make here. |

| | |
|---|---|
| Page #: 1
Line #: 29-30 | There's a problem with your sentence construction here: "The SAR receives echoes … received by the same instrument…". Revise this sentence for clarity. |
| Page #: 2
Line #: 32-34 | I don't think your description of SAR here is capturing the salient idea. I'm not sure what it means to "vary the number of radar echo locations within a track" -- it sounds like you are varying the number of targets? The constant along-track resolution is achieved by collecting returns from the same target at a large number of instrument positions. I would rephrase these two sentences for clarity. |
| Page #: 2
Line #: 36 | "Reflectivity" is an inherent property of an interface, it doesn't depend on the platform. You might argue that the measured echo strength could vary depending on the platform, but I think that is imprecise -- the same interface measured by a radar on a Twin Otter and a Basler flying at the same elevation should have the same apparent reflection strength. I would simply cut this sentence. |
| Page #: 2
Line #: 48-49 | I'm not sure what you mean by this sentence -- for example, what does it mean for there to be diversity in angular resolution? I would rephrase to clarify your key idea here. |
| Page #: 2
Line #: 54-55 | I think the word-choice in this sentence obscures the intended meaning. "… to facilitate location in data repositories and subsequent processing …", specifically, is awkward. After all, what does it mean to "locate data in processing"? I would rephrase to something much simpler: "We describe the data using trace numbers to more easily enable replication of our work." |
| Page #: 4
Line #: 69-70 | This sentence is really important, as it underlies the method, but I think the phrasing is unclear. For example, what is the "it" in the final clause? I would consider rephrasing to improve clarity. |
| Page #: 4
Line #: 72 | This should be (Heister and Scheiber, 2018), as it is referred to elsewhere in the manuscript. |
| Page #: 4
Line #: 82 | Can you help me understand how anisotropy factors in here? |
| Page #: 5
Line #: 94-95 | Can you clarify what you mean when you say "The spectrum can be divided into several sub-bands by a filter array (bank)"? |

| | |
|---|---|
| Page #: 8
Line #: 160-161 | I find the sentiment of this sentence somewhat confusing -- the idea that you would use the doppler centroids to generate the image, and then use the image to identify the dip direction. The image provides a nice qualitative representation of the range-Doppler domain data, but it is in the underlying spectra that the quantitative information lies, and any automated analysis would start there. |
| Page #: 9
Line #: 172 | I think there is a noun missing in this sentence before "are interpreted as". |
| Page #: 12
Line #: 203 | What do you mean when you say "the location of which are unexplained." The location of the sidelobes? |
| Page #: 13
Line #: 234-235 | This idea of differening resolution is interesting -- could you build that out a bit more for the reader? |
| Page #: 14
Line #: 251-255 | I think this is great! This highlights to whom this product is likely to be the most useful. The more you can reach the community working on SAR focusing with this work, the higher its impact will be. |

---

## Author Response (AR1)

Dear Editor and Referees,

We very much appreciate the time and effort you have committed to helping us revise this manuscript. We would like to acknowledge that this input has significantly improved the article and resulted in an accessible manuscript of broad interest to the cryosphere community that we are now confident will reach a wider audience.

Below we provide our corrections and responses to your comments, outlining the article modifications or additions. Following reviewer comments **in bold**, our responses are in black and any new text in *blue italics*.

**Referee #1**

Journal selection - Given the positive review we assume that your rejection of the article refers to your suggestion to submit to IEEE rather than The Cryosphere. We appreciate your acknowledgement of the wider value of the method and agree that it will have broad application. However, our preference is to continue with publication in The Cryosphere where it will be of direct interest to the readership. IEEE is a technical journal that many glaciologists do not have access to through institutional subscriptions (including those at the British Antarctic Survey, for example). We therefore do not believe that publication in IEEE would reach the widest glaciological audience, which is our intention. Also, we know of researchers in the cryosphere community who are already applying the method to their own data sets, and so there is an urgency to finalise this manuscript here to support those ongoing research efforts. We are also collaborating with colleagues outside of the cryosphere community who are applying the method and are therefore confident it will also reach a broad audience. Finally, as The Cryosphere is an open-access publication, it will also be available to the entire research community.

**1) Page 1; Line 11-12**

The two sentences starting "Due to this averaging..." are unclear to me. What is a "directional feature"? What is the "distance rate from the radar"? Consider revising these for clarity.

Apologies for the confusing text. We changed the two sentences to:

"Synthetic Aperture Radar (SAR) is a processing technique based on averaging the radar echoes received at multiple locations as the radar moves relative to the target. Due to this averaging, the directional properties of the back-scattering from the target received at these multiple locations are lost. We introduce an alternative representation of SAR images that preserves directional information encoded in its Doppler spectrum: the Doppler frequency accounts for the time-delay variation from the radar to the target."

**2) Page 1; Line 24**

Your verb choice here makes me stumble as I read -- what does it mean to use radar to "address processes and features"?

We will change the word "address" to "observe":

"... , SAR systems are used to observe processes and features at the surface, base and within the ice column."

**3) Page 1; Line 28**

What do you mean when you say that coverage "depends on the trajectory of the moving platform"? Why specifically "the trajectory"? I'm not sure what point you are trying to make here.

With trajectory we meant how densely the area of the survey was gridded. We changed the sentence to:

"SAR provides broad areal coverage, dependent on the platform carrying the radar."

**4) Page 1; Line 29-30**

There's a problem with your sentence construction here: "The SAR receives echoes ... received by the same instrument...". Revise this sentence for clarity.

We changed the sentence:

"In active mode, the SAR both transmits and receives the signal, whereas in passive mode, the SAR receives a signal from a different source"

**5) Page 2; Line 32-34**

I don't think your description of SAR here is capturing the salient idea. I'm not sure what it means to "vary the number of radar echo locations within a track" -- it sounds like you are varying the number of targets? The constant along-track resolution is achieved by collecting returns from the same target at a large number of instrument positions. I would rephrase these two sentences for clarity.

We changed the sentence to

"This is achieved by increasing the number of radar locations (i.e., instrument positions) used to focus deeper targets."

**6) Page 2; Line 36**

"Reflectivity" is an inherent property of an interface, it doesn't depend on the platform. You might argue that the measured echo strength could vary depending on the platform, but I think that is imprecise -- the same interface measured by a radar on a Twin Otter and a Basler flying at the same elevation should have the same apparent reflection strength. I would simply cut this sentence.

We agree and remove the sentence. We meant that it would depend on being a ground, airborne or spaceborne platform.

**7) Page 2; Line 48-49**

I'm not sure what you mean by this sentence -- for example, what does it mean for there to be diversity in angular resolution? I would rephrase to clarify your key idea here.

The antennas under the wing have a different separation to the antennas under the belly and are also offset vertically. The across-track angular resolution and beamwidth from each sub-array depend on the antenna separation. We rephrased to

"An array of 12 antennas is mounted on a Twin Otter aircraft, with a non-linear and non-uniform distribution imposed by the aircraft structure. This antenna distribution makes PASIN2 an array with diversity in three joint domains: spatial distribution, across-track angular orientation (due to the sub-array orientations), and across-track angular resolution (due to the distance between the sub-array elements)."

**8) Page 2; Line 54-55**

I think the word-choice in this sentence obscures the intended meaning. "... to facilitate location in data repositories and subsequent processing ...", specifically, is awkward. After all, what does it mean to "locate data in processing"? I would rephrase to something much simpler: "We describe the data using trace numbers to more easily enable replication of our work."

We agree. We meant to facilitate the processing. Now it is:

"We employ trace numbers instead of distance to more easily enable replication of our work, as distance..."

**9) Page 4; Line 69-70**

This sentence is really important, as it underlies the method, but I think the phrasing is unclear. For example, what is the "it" in the final clause? I would consider rephrasing to improve clarity.

We substitute the pronoun "it" by "the radar moves in ...", and the sentence becomes "The Doppler frequencies represent the rate variation of the range delay from the radar to each scatterer as the radar moves in the along-track dimension."

**10) Page 4; Line 72**

This should be (Heister and Scheiber, 2018), as it is referred to elsewhere in the manuscript.

Yes, we correct as you suggest.

**11) Page 4; Line 82**

**Can you help me understand how anisotropy factors in here?**

With anisotropy we meant the directional variation in how the target echoes are backscattered. When we focus the SAR images we typically assume isotropy, so we include all the samples within the SAR aperture; but in the specular layers we have a narrower beamwidth. To clarify, we include "non-uniform" next to "anisotropic", and substitute "spatial variability" by "directional variability". Now the text is

"Regardless of the algorithm used, focussing involves an integration, or average, after which only the output (average) is presented, thus omitting the directional variability of the input data. This variability is directly related to the anisotropic (non-uniform) angular backscattering within the incidence beamwidth, and therefore of importance at the interpretation stage."

**12) Page 5; Line 94-95**

Can you clarify what you mean when you say "The spectrum can be divided into several sub-bands by a filter array (bank)"?

With "array (bank)" we meant a group of filters. In digital processing "bank" is sometimes used, but we now remove it.

**13) Page 8; Line 160-161**

I find the sentiment of this sentence somewhat confusing -- the idea that you would use the doppler centroids to generate the image, and then use the image to identify the dip direction. The image provides a nice qualitative representation of the range-Doppler domain data, but it is in the underlying spectra that the quantitative information lies, and any automated analysis would start there.

We agree with your comment, and your interpretation is correct, we have therefore reworded the sentence accordingly. For a quick visual interpretation of the coloured image, we agree on it being more qualitative. The underlying data (RGB Bands) are used for the quantitative analyses that we suggest, like layer tracking and focussing quality (both mentioned in the article), and calibration of antennas (not mentioned).

"In the RGB-DD image (Fig. 3b), the steep (dark) layers are now represented by a colour, reflecting the underlying pattern in the Doppler spectra, highlighting features that are lost with standard SAR processing and images, and enabling a qualitative interpretation. The underlying data, i.e., the Doppler bands (RGB-DD output), may be analysed in a quantitative and automated manner to extract features such as surface slope and dip direction. Furthermore, with the three RGB-DD filters, different decompositions may be achieved by varying the centroids and bandwidths and combining the results to produce further bands, like those in Fig. 3d, allowing additional quantitative analysis and improved angular resolution."

**14) Page 9; Line 172**

I think there is a noun missing in this sentence before "are interpreted as".

Yes, you are right. Now the sentence is

"... as crevasses behave as corner reflectors backscattering the EM waves in all directions, these bright-white features are interpreted as crevassing extending towards the surface near the margin."

**15) Page 12; Line 203**

What do you mean when you say "the location of which are unexplained." The location of the sidelobes?

Yes, the range location. Now the text is

"... is produced by range sidelobes of -20 dB (replicas of the tephra feature); the range location of these sidelobes is currently unexplained."

**16) Page 13; Line 234-235**

**This idea of differing resolution is interesting -- could you build that out a bit more for the reader?**

To estimate the slopes, we want finer beamwidth. But broader beamwidth is used in the SAR image to improve the along-track resolution, the wider the beamwidth the finer the resolution, so there is a trade-off. We modified the sentences, to be

"The main advantage of the multi-band method over RGB-DD is the enhanced angular resolution, which is five times better due to the 2° beamwidth compared to the 10° beamwidth of the RGB method. Conversely, the along-track spatial resolution of the RGB-DD method is five times finer than with the multi-band method due to each of the RGD-DD filters integrating a wider beamwidth."

**Referee #2,**

1) I. 66-67: "the range dimension, related to the distance from the radar to the target of interest": I would suggest referring to the time delay rather than distance, as you do in the following, more detailed explanation (particularly relevant for sounding radars)

We agree. Now the text is

"... the time-delay (range delay) dimension, related to the distance from the radar to the target of interest..."

2) I. 114-115: Could be clarified that it is for a diffuse scatterer / rough surface (as opposed to the specular reflector discussed afterwards)

We modify the beginning of the paragraph, as

"If a target with diffuse scattering (a rough surface) is perfectly focused, when..."

3) I. 117: Maybe specify that it is for a specular reflector

We moved the adjective "specular" almost to the beginning of the sentence and modify the sentence structure, to emphasize is for this type of scatterers. The text now is "When a specular (mirror like) target is focused, only a narrow bandwidth is backscattered (Fig. 2d, e), and its RGB colour will be..."

4) Figure 4 (d): Could you clarify whether the surface slope is projected on the focusing plane, or is it the slope relative to the vertical, independent of the direction?

Our apologies for the confusion. Because the text only mentions the altimeter we assumed it would be interpreted as elevation, whereas the caption mentions the slope. This is the surface slope from the horizontal. In the caption of Fig. 4d we add

"(d) surface slope in the along-track dimension (reported in degrees from the horizontal), from LIDAR"

and in line 181 we modify the in-text sentence to be

"From the on-board Lidar altimeter, we calculate the ice surface slope (measured from the horizontal) in the along-track direction (Fig. 4d), which highlights...".